bioinformatics/computational biology

renal cell carcinoma, kidney, RGS5, data analysis, RNA-seq data, statistical analysis

**Author for correspondence:**
Leili Shahriyari
e-mail: lshahriyari@umass.edu

# RGS5 plays a significant role in renal cell carcinoma

## Sumeyye Su[1] and Leili Shahriyari[2]

[1]Department of Mathematics, University of Texas Arlington, Arlington, TX 76019, USA
[2]Department of Mathematics and Statistics, University of Massachusetts Amherst, Amherst, MA 01003, USA

 LS, 0000-0001-6234-8449

Recent advances in biotechnology led to generation of large complex biological and clinical datasets that can be used to infer the underlying mechanism of many diseases and arrive at personalized treatments. One of these datasets are the whole genome profiles, including a good collection of publicly available human gene expression datasets. In this project, we analysed gene expression profiles of patients with renal cell carcinoma (RCC). We found that the regulator of G-protein signalling 5 (RGS5) might play a crucial role in initiation and progression of RCC, and it might be prognostic. We observed that a high expression level of RGS5 is associated with better survival months. Importantly, when the grade of tumour increases, the RGS5 expression level significantly decreases. Although there is no difference between expression level of RGS5 in male and female patients with primary tumours in the right kidney, among patients with primary tumours in the left kidney, females have a significantly higher RGS5 expression than male patients. Interestingly, we also observed a significant association between the high expression level of RGS5 and low serum calcium level and elevated white blood cells level.

## 1. Introduction

Renal cell carcinoma (RCC) is the most commonly seen malignant tumour type in adult kidneys [1], and it is considered a morphologically and genetically heterogeneous tumour [2]. The incidence rate of RCC is around twice as common in males as in females [3,4], and the most common risk factors are obesity, diabetes and hypertension [5]. According to the US National Cancer Institute, there were approximately 533 204 people with kidney and renal pelvis cancer in 2016, and 73 820 people are estimated as a new case in 2019 which is 4.2% of all new cancer cases.

Although 74.8% of the patients survived 5 years or more in 2009–2015, 14 770 people are estimated to die in 2019 because of this disease [6].

**Figure 1.** (*a*) Shows the average expression levels of each gene across patients by blue dots and the corresponding average plus standard deviation by orange dots. (*b*) Shows the average of gene expression profile of a primary tumour by blue dots, and the corresponding average plus standard deviation by orange dots.

To understand the process of initiation and progression of RCC and discover effective treatments, several datasets, including clinical and pathological information, genomic alterations, DNA methylation profiles, and RNA and proteomic profiles of RCC primary tumours, have been collected and analysed. For example, by analysing these datasets, The Cancer Genome Atlas (TCGA) research network found a correlation between worsened prognosis in patients with the most common type of RCC, clear cell renal cell carcinoma (ccRCC), and a metabolic shift involving increased dependence on the pentose phosphate shunt, decreased AMPK, decreased Krebs cycle activity, increased glutamine transport and fatty acid production [7]. Additionally, a study of 103 RCC patients showed a significantly high FABP7 mRNA expression in men and a high expression of BRN2 protein in women. They have also observed a poor prognosis in females with low FABP7 and high BRN2 expression [8].

In this project, we analysed the RNA-seq data of the primary tumours of 534 patients with RCC to see if there is any signature of clinical or demographic information of the patients in the gene expression data of their primary tumours. We collected the data of the TCGA project from cBioPortal; tables in figures 2–5, 7 and 8 show an overview of demographic and clinical features of the patients. These tables provide the number of patients in each subcategory, including gender, type of tumour, survival status, etc. Differences in the numbers are due to missing information for some patients.

The first crucial step in analysing datasets is the normalization, which has a significant effect on the results. One of the most common normalization methods for gene expression data is the standardization of the values of each gene. However, it has been shown that this routine approach might not be a good one [9]. By analysing the gene expression profiles of colon primary tumours, Shahriyari showed that the distribution of expression levels of genes across patients are very different from one gene to another one, while the distribution of gene expression levels of one patient is very similar to another patient. We have also observed the same thing in RCC. Figure 1*a* shows the average value of each gene and their standard deviations; as one can see the average and standard deviation of expression levels might be hugely different from one gene to another gene. However, the average and standard deviation of gene expression levels of patients are very similar (figure 1*b*). In other words, the distribution of expression levels of genes in one patient is very similar to another patient. Note, the standardization destroys this crucial statistical information of the data. For this reason, we scale the gene values of each patient separately to get the relative amount of each gene in each primary tumour, for more details see the Methods section.

Another challenging step in analysing gene expression datasets is dimensionality reduction mainly because the number of genes is much higher than the number of samples. The situation is worse for human datasets compared with animals or plants because of the lack of number of samples and existence of large number of genes. Here, we used variance threshold methods to get top variant genes. After normalization, we calculated the standard deviation of each gene across samples. We found that the top 10 most variant genes across patients are B2M, CD74, EEF1A1L14, GAPDH, GPX3, IGFBP3, RGS5, SPARC, TGFB1 and VIM. Among these genes, RGS5 looks very interesting (see the following results).

We found that RGS5 gene expression might be prognostic and might have a significant role in initiation and progression of RCC. The regulator of G-protein signalling (RGS) family regulates cellular signalling events downstream of G-protein coupled receptors (GPCRs), which have been found to be associated with the initiation and progression of multiple cancers [10]. A high expression level of RGS5, a member of RGS family, has been detected in various human tissues such as heart, skeletal muscle, bladder, uterus, alimentary tract and HUVECs [11–13]. RGS5 is also involved in ion transport mechanisms in the kidney and has been associated with the regulation of blood pressure or hypertension [14].

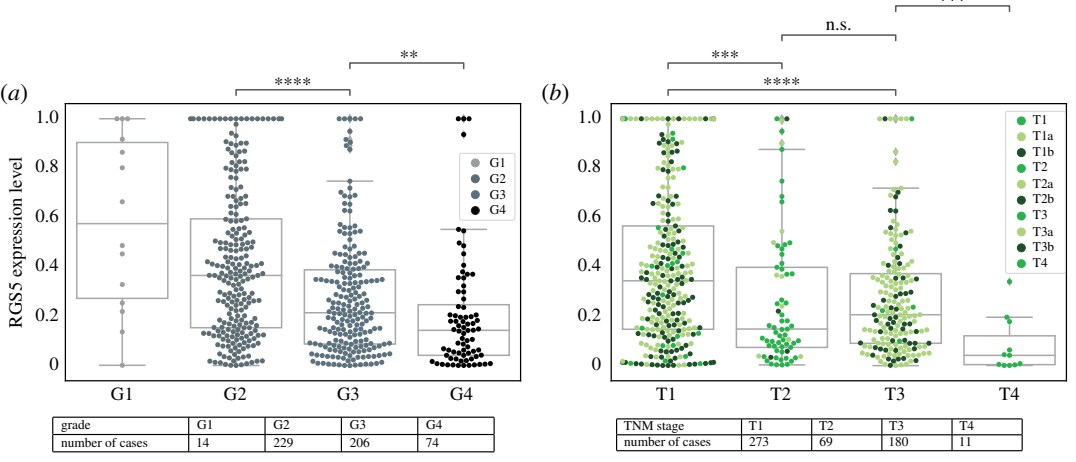

**Figure 2.** Expression level of the RGS5 gene as a function of grade and TNM staging of RCC. (*a*) Shows that the expression level of RGS5 in the patients' primary tumour decreases when the grade of tumour increases. (*b*) Represents the expression level of RGS5 in primary tumours categorized based on TNM stage of RCC. For some patients with the stage *Ti* (green), *i* = 1, 2, 3, 4 cancer, we know if the patient had the stage *Tia* (light green) or *Tib* (dark green) cancer, for more information, see table 1. Tables in this figure indicate the number of patients in each category.

In a study using reverse transcription–polymerase chain reaction (RT-PCR), which is a strongly sensitive method for the detection and amplification of mRNA, a very weak or undetectable expression level of RGS5 has been observed in normal kidneys, while a high level of RGS5 expression has been detected in all RCCs [15]. Importantly, the same study indicates that the tumour endothelial cells are the main location of RGS5 in RCC [15]. Moreover, a study on DNA microarray data of 27 RCCs using multivariate Cox analysis suggested that vascular cell adhesion molecule 1 (VCAM1), endothelin receptor type B (EDNRB) and RGS5 are predictors of survival months [16].

On the other hand, a strong association has been observed between high RGS5 expression level and better survival in 51 non-small cell lung cancer (NSCLC) patients [17]. They detected a high RGS5 expression in 47% of NSCLC patients, and low expression of RGS5 has been observed to be linked with cancer vasculature invasion and lymph node metastasis [17]. In another study of 127 human paraffin-embedded epithelia ovarian cancer (EOC) tissue samples, RGS5 gene expression in EOC tissues was higher compared with normal ovaries, and EOC patients with high RGS5 expression had a better survival with progression-free [18]. In contrast to these two study, it has been found that over-expression of RGS5 gene in the human lung cancer cells decreases the survival rate, and increases the cytotoxic result of radiation [19].

# 2. Results

## 2.1. Expression level of RGS5 significantly decreases when the grade of tumour increases

We found that the grade of RCC tumours is a decreasing function of the normalized value of RGS5 (figure 2). The grades of tumours, which are determined by the appearance of the cancer cells under the microscope, provide us with some insight about how the cancer might behave. Lower grades represent the cancer cells that look more like normal cells; G1 and G2 tumours tend to grow slowly and spread less. Higher grades represent the cancer cells that look more different from the normal cells. Cancer cells in G3 and G4 tend to grow quickly and spread fast [20]. We found that there is a significant reduction in the RGS5 expression from the G2 to the G4 grade. Note, we cannot make a firm conclusion for grade one tumours, because there are not many G1 patients.

## 2.2. Stage1 tumours have the highest expression level of RGS5 compared with the other stages

In figure 2, we separated the tumour stage of patients by using the most commonly used staging system, TNM, for kidney cancer (table 1) [21]. Since there are not many patients with stage 2 and 4 cancer, one may ignore the T2–T3 and T3–T4 differences and conclude a significant decrease in the RGS5 expression level from T1 to T3 (figure 2).

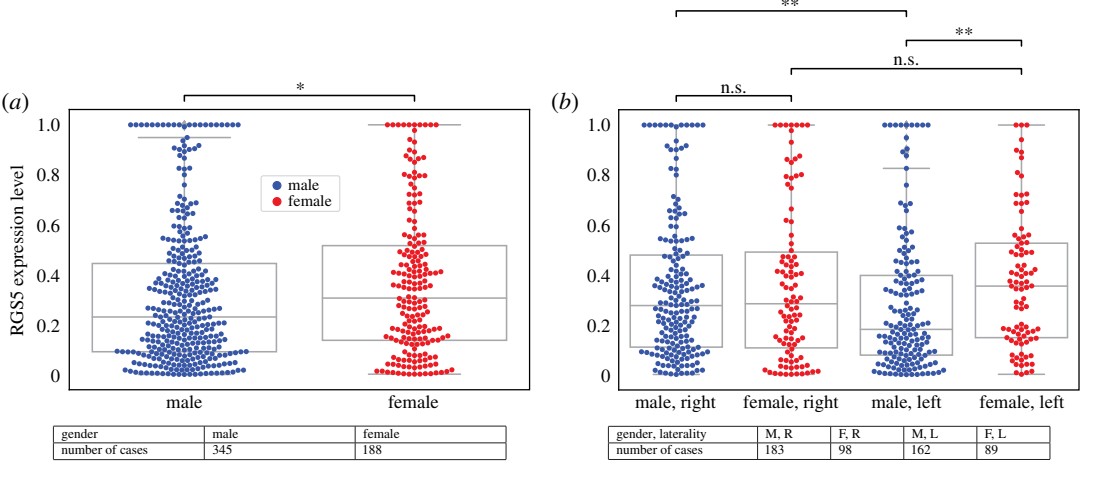

**Figure 3.** Expression level of RGS5 as a function of gender and the location of the primary tumour. (*a*) Shows that female patients have higher expression level of RGS5 than male patients. Grouping patients based on the location of their primary tumours, (*b*) shows that the difference between the expression level of RGS5 in female and male patients is due to the fact that male patients with primary tumours in the left kidney have lower expression level of RGS5 than other patients. Tables in this figure indicate the number of patients in each category.

**Table 1.** TNM staging for kidney cancer.

| TNM | disease status | size |
| --- | --- | --- |
| T1 | tumour limited to kidney | less or equal to 7 cm |
| T1a | tumour limited to kidney | less than 4 cm |
| T1b | tumour limited to kidney | more than 4 cm |
| T2 | tumour limited to kidney | more than 7 cm |
| T2a | tumour limited to kidney | between 7 and 10 cm |
| T2b | tumour limited to kidney | more than 10 cm |
| T3 | growing into the fat around the kidney | |
| T3a | tumour invades renal vein or fat but not beyond Gerota's fascia | |
| T3b | tumour extends to vena cava below the diaphragm | |
| T4 | tumour invades beyond Gerota's fascia | |

## 2.3. Tumours originated in the left kidney have a significantly higher RGS5 expression in females than male patients

We observed a significant high RGS5 expression in female patients versus male patients. However, further investigation revealed no differences in the expression level of RGS5 in males and females for tumours that originated in the right kidney. Inevitably, there is a significant difference in RGS5 expression between female and male patients with tumours in the left kidney. Importantly, we found that male patients with a primary tumour in the right kidney have a higher level of the RGS5 expression compared with the left kidney. However, there is no significant difference between RGS5 expression of the female patients with a primary tumour in the left kidney compared with the right kidney (figure 3).

## 2.4. Tumour-free patients have a significantly higher level of RGS5 expression versus patients with tumours

We further investigated the relationship between RGS5 expression in primary tumours and the patients' tumour status, which provide us with the state or condition of individuals' neoplasm at the follow-up

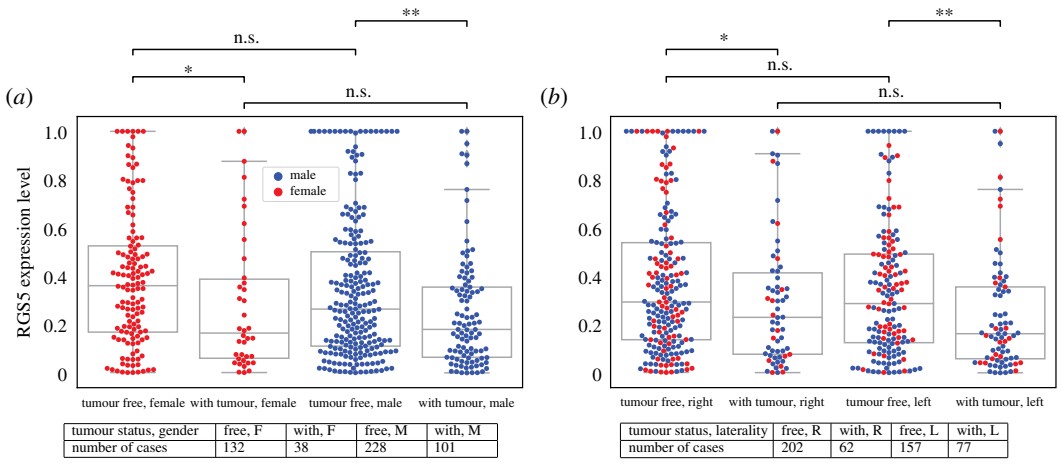

**Figure 4.** Expression level of RGS5 gene in primary tumours as a function of tumour status at the time of last follow-up. This figure indicates that the primary tumours of RCC patients with no tumours at the time of last follow-up had a higher RGS5 expression level than patients with tumours at the time of last follow-up regardless of patients' gender (*a*) and the location of primary tumours (*b*). Tables indicate the number of patients in each category.

time. If patients did not have any tumour at the last time of follow-up, they were categorized as 'tumour free', and if they had any tumour then they were marked as 'with tumour'. We observed a higher level of RGS5 expression in tumour-free patients compared with patients with tumour. When we divided kidney cancer patients into subgroups based on their gender, location of the primary tumour and tumour status, the gene expression level reacts similarly to general results; aggressive tumours have a significantly lower expression level of RGS5 (figure 4).

## 2.5. RGS5 expression of living patients is higher than that of deceased patients

When we consider overall patients' survival status, we see from figure 5 that RGS5 expression level of living patients is higher than deceased patients. Also, when we divide the patients into subcategories based on gender, results are not affected; so living patients, regardless of their gender and even white blood cell levels, have a higher level of RGS5 expression compared with deceased patients. In addition, when we compared only living patients by their gender, we found that female living patients have a higher level of RGS5 expression compared with male living patients. Furthermore, the laterality does not affect the RGS5 expression level as much as the survival status (figure 5). Moreover, we see that there is a similar tendency in the RGS5 expression with disease-free status and overall survival status, as expected. However, there is no correlation between age at diagnosis and RGS5 expression level.

## 2.6. Patients with higher level of RGS5 have higher survival months

We observed no correlation between the level of RGS5 and overall survival months and also age at diagnosis (figure 6). However, when we divided RGS5 expression level into two categories of low (patients with normalized RGS5 ≤0.1) and high (patients with normalized RGS5 >0.1), we observed that the patients with a high RGS5 expression have better survival compared with the patients with low RGS5 expression. This result aligns with our other observations regarding the grade of tumours (figure 2), tumour status (figure 4) and survival status (figure 5), as we expected.

## 2.7. The expression of RGS5 in patients with low serum calcium levels is significantly higher than those with normal levels

It has been observed that kidney disorders are highly correlated with low calcium levels in the blood (hypocalcaemia) [22]. We found that the patients with low calcium levels have a higher RGS5 expression level compared with the patients with normal calcium levels. In addition, considering laterality, we observed that primary tumours in both right and left kidneys follow the same trend,

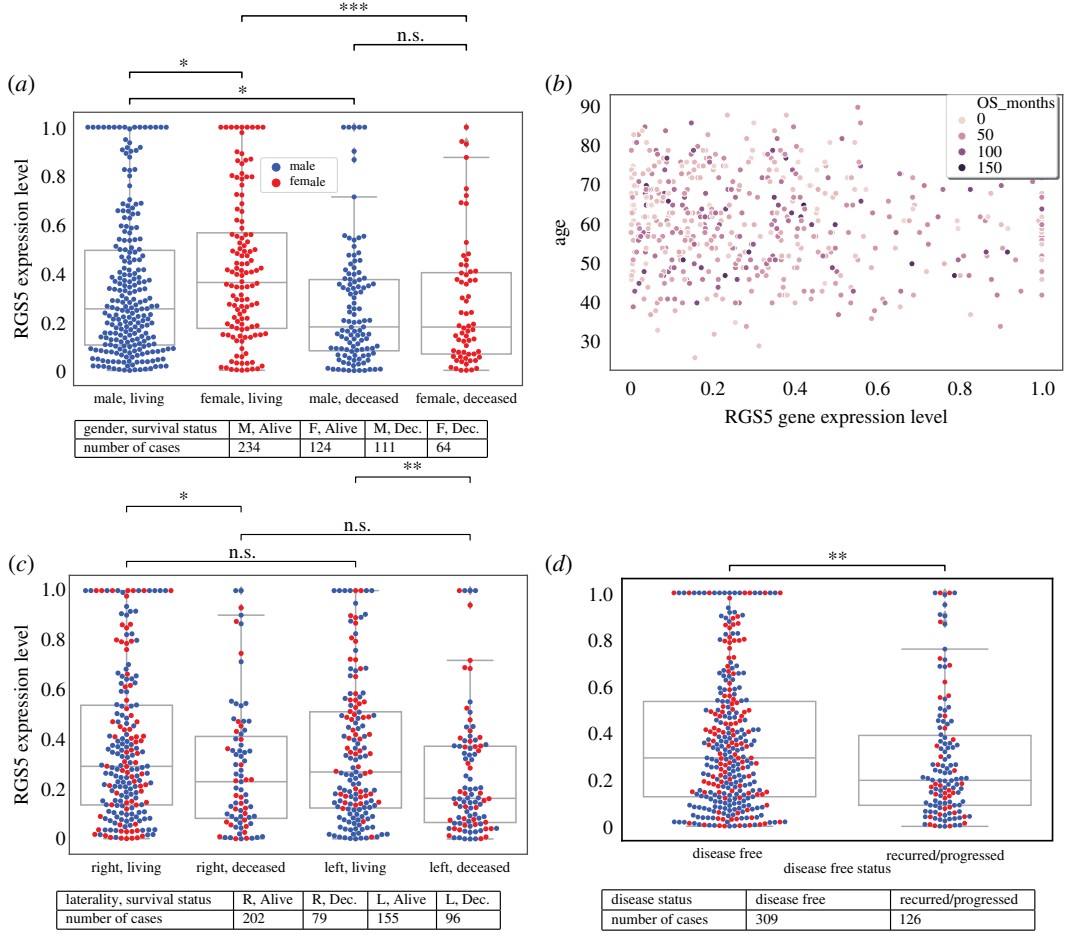

**Figure 5.** Expression level of the RGS5 in primary tumours as a function of survival status of patients at the time of last follow-up. The left subplots (*a*,*c*) show that primary tumours of alive patients have a higher expression level of RGS5 compared with the deceased patients regardless of gender (*a*) and the location of primary tumour (*c*). (*b*) Indicates that there is no linear correlation between age at diagnosis and RGS5 expression. (*d*) Shows that primary tumours of patients, who were diseased free at the last time of follow-up, have a higher expression level of RGS5 compared with the patients with reoccurred cancer. Tables indicate the number of patients in each category. In all subfigures, female patients are represented by red colour and males by blue.

although the difference is not significant in the left kidney. Similarly, when we grouped patients based on their gender, the RGS5 expression of male patients who have low calcium levels is significantly higher than those with normal calcium levels, but it is not significant for female patients. Lastly, when we divided patients into subcategories based on overall survival status, we still observed a higher level of RGS5 in living patients compared with the diseased ones regardless of their calcium level. Importantly, deceased patients with normal calcium levels have the least level of RGS5 expression. Note, the percentage of living patients with low serum calcium levels ($138/(138 + 66) \approx 0.68$) is higher than those with normal levels ($89/(89 + 61) \approx 0.59$) (figure 7).

## 2.8. Patients with elevated white blood cells have a high level of RGS5 expression

Elevated white blood cells (WBC) is known as a common predictor of chronic kidney disease (CKD) [23]. We observed that the RGS5 expression in the patients who have elevated WBC was greater than those in the patients who have normal WBC. In addition, when we considered the tumour status of the patients, again tumour-free patients have a higher RGS5 expression level than with-tumour patients regardless of their WBC counts. Furthermore, the percentage of living patients with elevated WBC ($118/(118 + 46) \approx 0.72$) is higher than those with normal levels ($164/(164 + 103) \approx 0.61$). Lastly, when we divided the patients into subcategories based on serum calcium level and WBC level, we observed that calcium level is more important than WBC for predicting the expression level of RGS5 (figure 8).

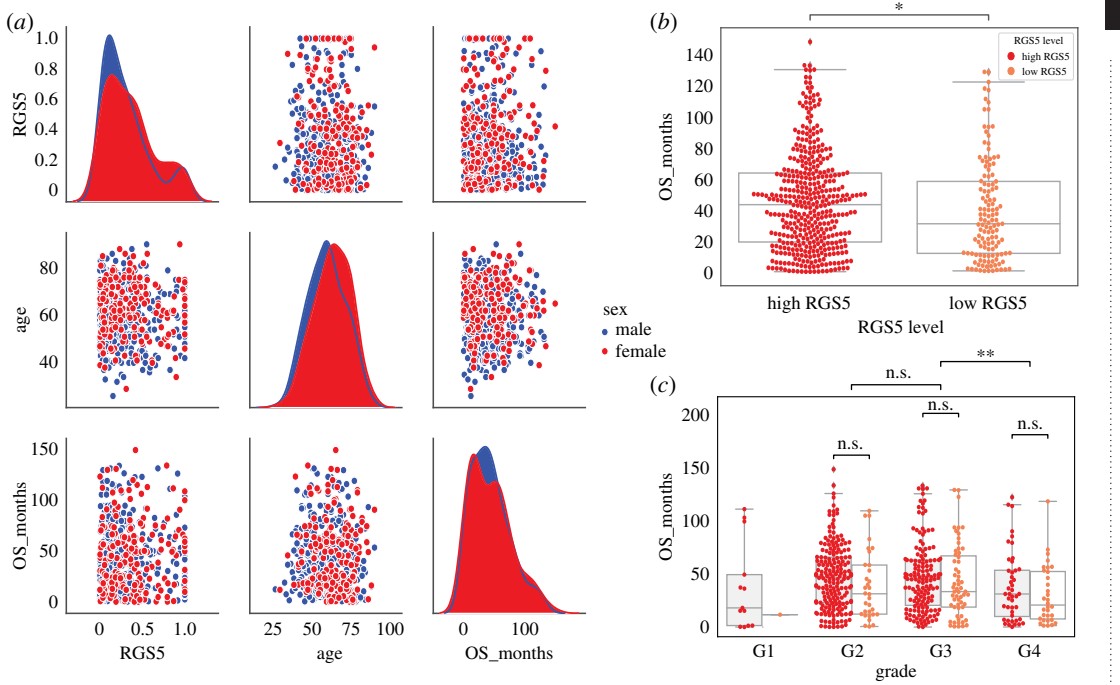

**Figure 6.** Overall survival months as a function of the RGS5 expression level. (*a*) Represents the values and distributions of RGS5 expression level, age at diagnosis, and survivals months for female and male patients. (*b,c*) Show the difference between survival months of patients with low (normalized RGS5 expression value less than or equal to 0.1) and high (greater than 0.1) RGS5 expression levels. In (*c*), patients are grouped based on the grade of their tumours.

## 2.9. Expression level of RGS5 might be prognostic

To sum up, figure 9 shows a hierarchically clustered heat-map of the above-mentioned clinical and demographic features with the correlation metric. As a result, RGS5, survival months and survival status ended up in the same cluster, and then they grouped with WBC level and laterality. As one could expect grade and stage of the tumour are clustered with the tumour status and the disease status, and interestingly they all clustered with serum calcium level.

# 3. Discussion

Recent advances in biotechnology have led to the development of new immunotherapeutic methods for treating various cancer types. One of the crucial requirements of these methods is explaining structure of tumour-specific antigens (TSA) or tumour-associated antigens (TAA) that help to fight with malignant cells. Krüger *et al.* identified RGS5 as one of the RCC-associated antigens [24]. Boss *et al.* also found a significant upregulation of RGS5 in a broad variety of malignant cells, and by identifying two HLA-A2- and HLA-A3-binding peptides derived from the RGS5 protein, they suggested RGS5 peptides as candidates for designing cancer vaccines to target malignant cells and tumour vessels [25].

RGS family are known as signal transaction molecules that are associated with the arrangement of heterotrimeric G proteins by acting as GTPase activators. Moreover, RGS5 is a hypoxia-inducible factor-1-dependent involved in the induction of endothelial apoptosis [26]. Furthermore, it has been shown that the variants in RGS5, ATP1B1 and SELE genes accounted for 2–5 mm Hg differences in mean systolic blood pressure levels, and the cumulative effect reaches 8–10 mm Hg [27]. Hence, RGS5 is an important contributor to high blood pressure [28], and hypertension is one of the most common risk factors of renal disease [29]. There is also evidence that RGS5 can act as a physiological regulator of calcium sensing in the parathyroid gland as a inhibitor of calcium-sensing receptor (CaSR) signalling [30], and a study of patients with the stage 3–4 CKD determined that the lower serum calcium is associated with the higher risk of renal replacement therapy and rapid renal function progression [31]. Furthermore, a higher incidence rate of early mortality in CKD patients with low blood calcium levels has been observed [32]. Here, we noted a significantly higher expression level of

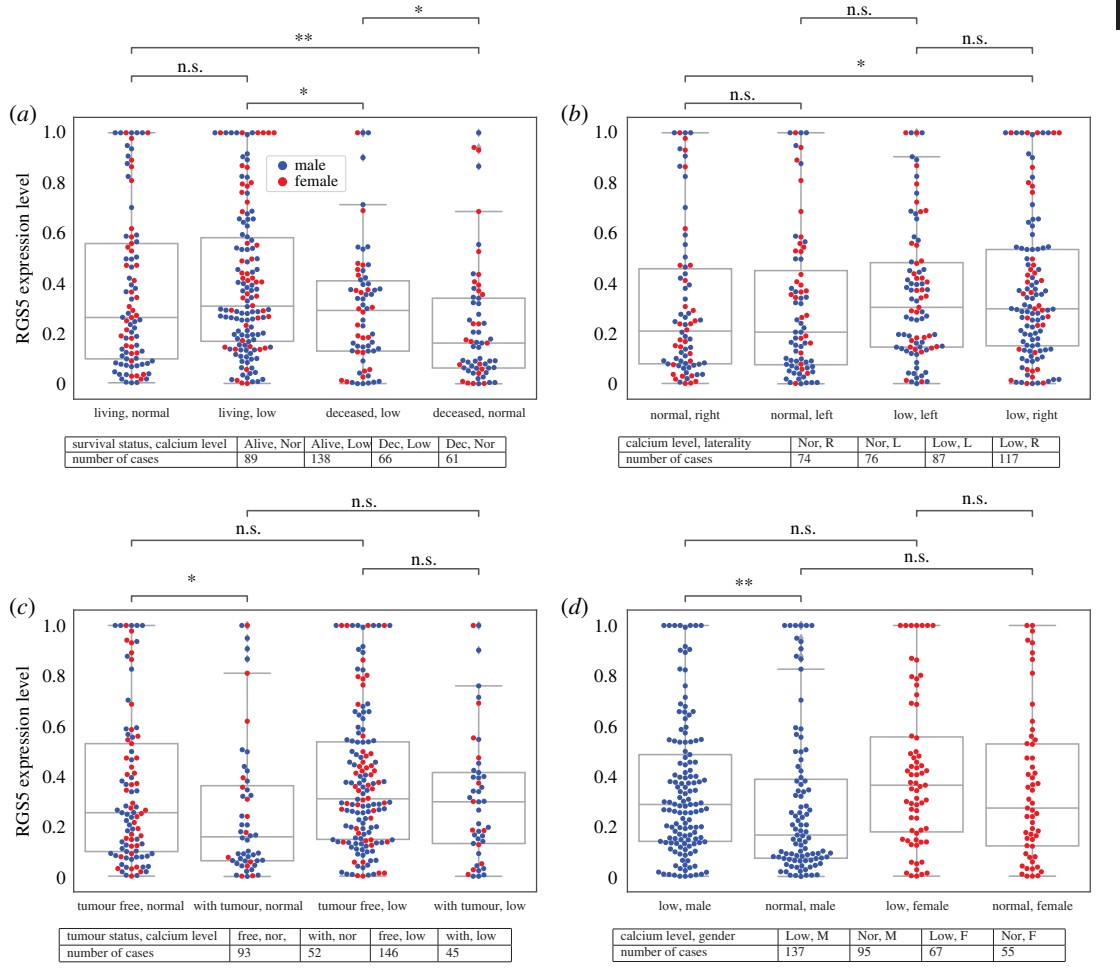

**Figure 7.** Expression level of the RGS5 gene as a function of serum calcium level and living status (*a*), laterality (*b*), tumour status (*c*) and gender (*d*). This figure shows that primary tumour of patients with low calcium level have a higher expression level of RGS5 compared with the patients with normal calcium level. Tables indicate the number of patients in each category. In all subfigures, female patients are represented by red and male patients by blue.

RGS5 in primary tumours of patients with low serum calcium levels than those with normal calcium levels.

Additional to the various biological roles of RGS5, there are some contradictory observations about the prognostic behaviour of RGS5. For instance, an association between high expression levels of RGS5 and poor outcomes have been observed in a study of human lung cancer [19], while studies on non-small cell lung cancer [17] and epithelia ovarian cancer [18] reported a link between a high expression levels of RGS5 and better survival months. Here, by analysing the RNA seq data of primary tumours of 533 patients with RCC, we independently observed a strong association between high expression levels of RGS5 and better outcomes. We found that the RGS5 expression level significantly decreases when the grade of tumour increases. Moreover, when we considered TNM staging system which gives us information about the size and location of the tumour (table 1), we observed a significant decrease in the RGS5 expression level when the stage increases from T1 to T3.

Several studies show that men with RCC have higher stage and grade and ultimately worse overall survival than women with RCC [3,33]. We also observed that the survived female patients have a significantly higher expression level of RGS5 compared with other categories (figure 5). Moreover, some studies reported better clinical results such as low grade tumours and better survival in patients with right-sided RCC compared with the left-sided RCC [34,35], but our result shows no significant difference between the expression level of RGS5 in female versus male patients with primary tumours located in the right kidney. However, among patients with primary tumours in the left kidney, female patients have a significantly higher RGS5 expression level (figure 4).

There is a correlation between elevated WBC and kidney function deterioration [36], and a study of 362 advanced renal cell cancer patients showed that elevated WBC is associated with poor survival [37].

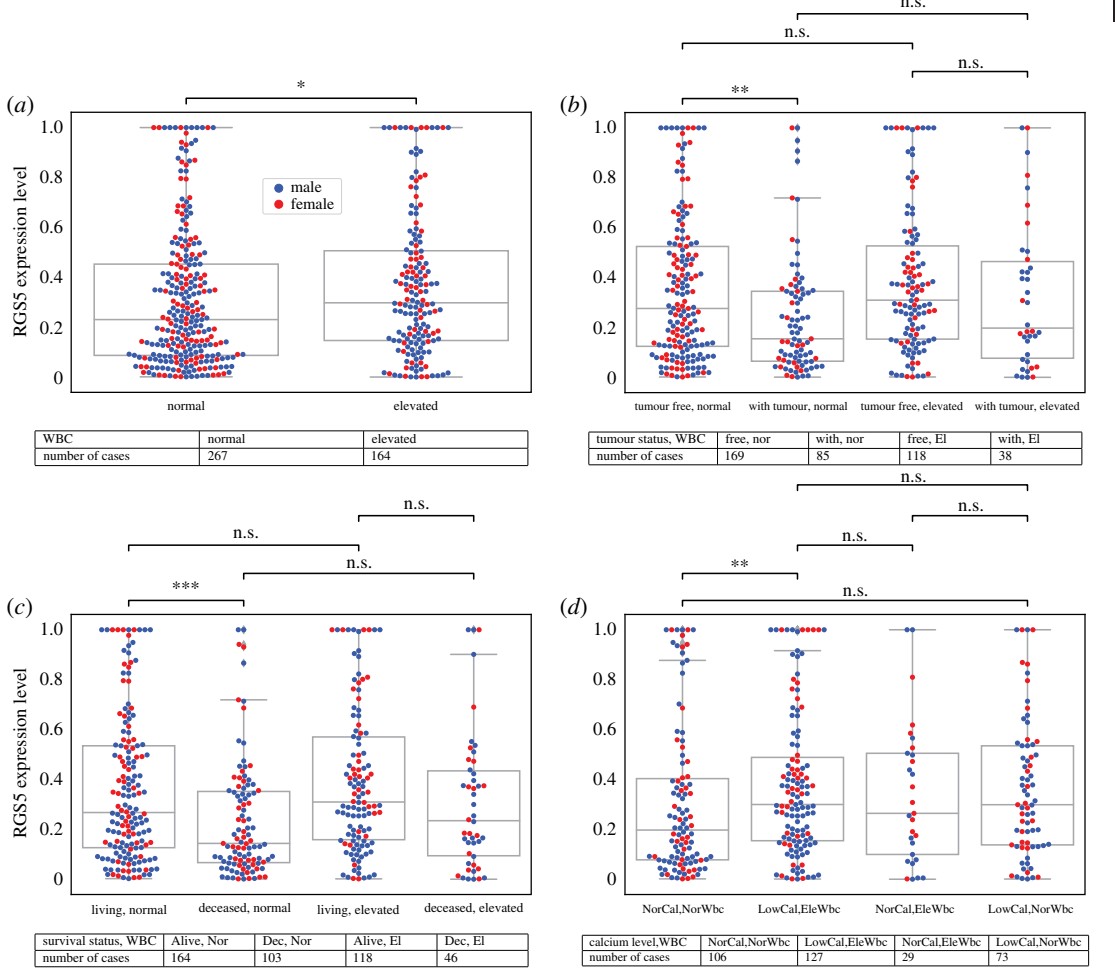

**Figure 8.** Expression level of the RGS5 gene as a function of WBC (*a*), WBC and tumour status (*b*), WBC and living status (*c*), and WBC and calcium level (*d*). Tables indicate the number of patients in each category. In all subfigures, female patients are represented by red colour and male patients by blue.

In our study, when we examined the relationship between RGS5 expression and WBC, we found that patients with elevated WBC have significantly higher RGS5 expression compared with those with normal WBC. Furthermore, we observed that the percentage of survived patients are higher in elevated WBC category than normal WBC group.

Although this study emphasizes that RGS5 plays a significant role in RCC, the different biological roles of RGS5 [38] and contradictory observations cause uncertainty about how RGS5 expression affects the outcome of treatments and how RGS5 could be used to develop better treatments. Therefore, future investigation is necessary to explore the role of RGS5 in initiation, progression, and suppression of RCC to arrive at effective treatments.

# 4. Methods

We downloaded the TCGA provisional data of patients with renal clear cell carcinoma (RCC) from the cBioPortal, and we divided the dataset into subcategories based on gender, tumour status, survival status, laterality, serum calcium level, white blood cell counts, grade and stage of tumour (figures 2–7).

In a recent study, it has been observed that patients have similar gene expression distributions, while the expression level of each gene across the patient has a different distribution [9]. Therefore, to avoid losing statistical information of the data, we normalized the gene expression datasets from the primary tumour by scaling gene expression of each patient separately [39]. Specifically, we have a dataset [$p_1, \ldots, p_n$], where $n$ is the number of patients. Each $p_i$ is a list of gene expression values, i.e.

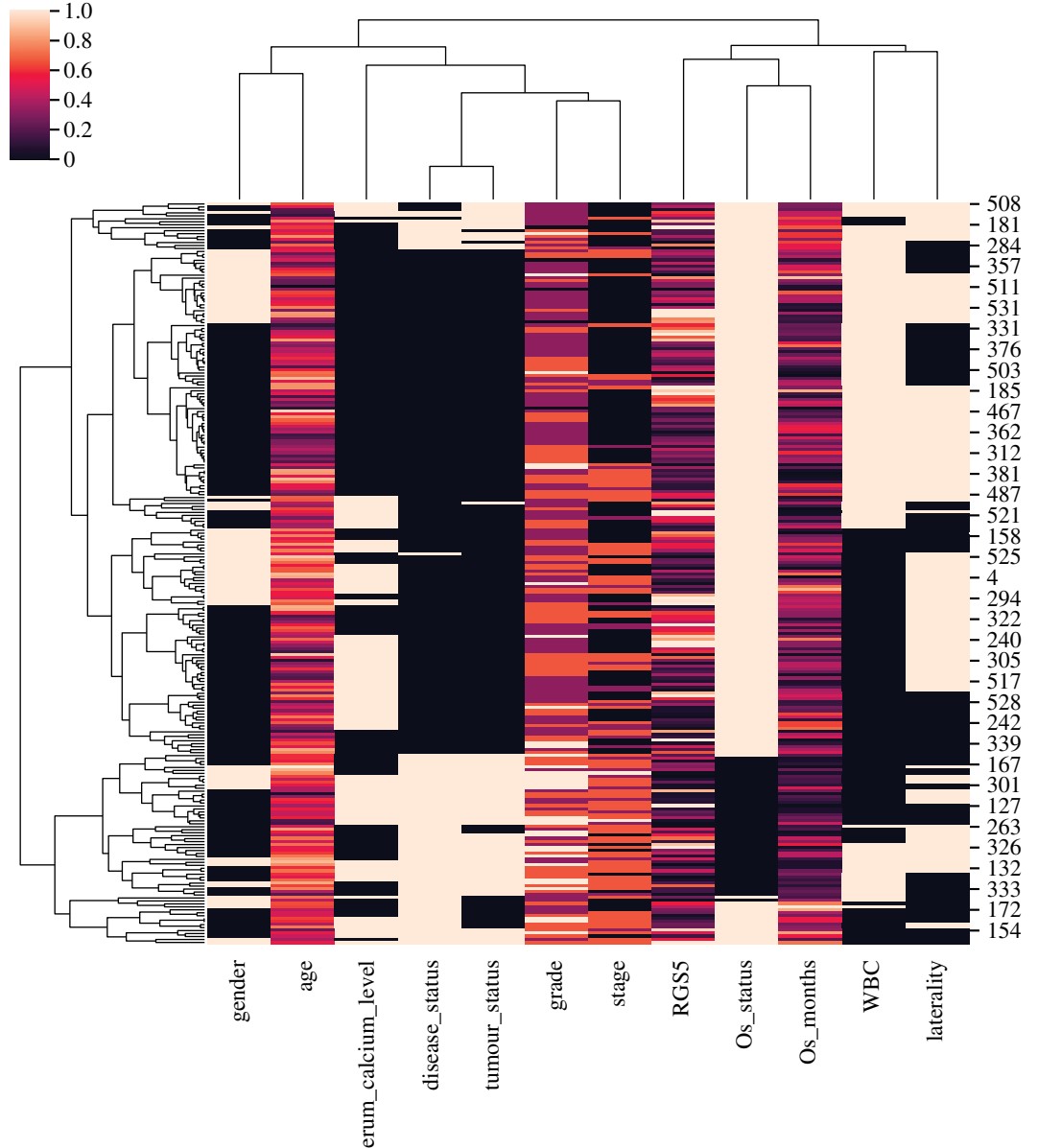

**Figure 9.** Hierarchically clustered heat-map of features that have been investigated in this paper and RGS5 gene. We converted the categorical features to numerical values, and then scaled all values between 0 and 1. Stage and grade of tumour are valued from 1 to 4. Gender: female: 0, male: 1. Tumour status: tumour free: 0, with tumour: 1. Laterality: left: 0, right: 1. Serum calcium level: low: 0, normal: 1. Disease status: disease free: 0, reoccurred: 1. OS status: living: 1, deceased: 0. WBC: normal: 0, elevated: 1. We then used correlation metric to cluster.

$p_i = [g_{i1} \dots, g_{im}]$, where each $g_j$ is the expression value of gene $j$, and $m$ is the total number of genes ($m =$ 20 531). That means the dataset is an $n \times m$-dimensional matrix $D = [g_{ij}]$, where $g_{ij}$ is the expression of gene $j$ in patient $i$. To use simple feature scaling normalization method, we found maximum gene expression value of each patient, and then we divided the values of each gene by the maximum gene expression values for the patient; i.e $q_i = max([g_{i1} \dots, g_{im}])$, $\hat{p}_i = [g_{i1}/q_i \dots, g_{im}/q_i]$, where $\hat{p}_i$ is the normalized gene expression vector for the patient. After that, we calculated the standard deviation of each gene across samples, and RGS5 was one of the top 10 genes with the highest variance.

For statistical analyses, we employed Mann–Whitney–Wilcoxon (MWW), also known as Mann–Whitney $U$-test. MWW is a non-parametric test of the null hypothesis that it is equally likely that a randomly selected value from one sample will be less than or greater than a randomly selected value from a second sample [40]. In figures, the significance levels have been shown by stars; n.s.: $0.05 < p \leq$ 1, *: $0.01 < p \leq 0.05$, **: $0.001 < p \leq 0.01$, ***: $0.0001 < p \leq 0.001$, ****: $p \leq 0.0001$.

Data accessibility. The codes are available on GitHub https://github.com/ShahriyariLab/RGS5. The RNA-seq and clinical datasets can be downloaded from https://www.cbioportal.org/study/summary?id=kirc_tcga [7,41].

Authors' contributions. S.S. carried out the study, performed data analysis and drafted the manuscript; L.S. designed and supervised the study, coordinated the study and helped draft the manuscript. All authors gave final approval for publication.

Competing interests. We declare we have no competing interests.

Funding. We received no funding for this study.

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
