## [Reviewer comments · Royal Society Open Science]

Review History

RSOS-191422.R0 (Original submission)

Review form: Reviewer 1

Is the manuscript scientifically sound in its present form?

Yes

Are the interpretations and conclusions justified by the results?

No

Is the language acceptable?

Yes

Do you have any ethical concerns with this paper?

No

Have you any concerns about statistical analyses in this paper?

Yes

Recommendation?

Major revision is needed (please make suggestions in comments)

Comments to the Author(s)

Summary :

In this work, the authors analyzed gene expression profiles of patients with renal cell carcinoma. They found that RGS5 might play a crucial role in initiation and progression of renal cell carcinoma. They have found that when the grade of tumor increases, the RGS5 expression level significantly decrease. They also found that high expression levels of RGS5 are associated with better survival amongst patients. The authors then further investigate this data when separating male and female patients as well as left versus right primary tumors. Finally, they observed a significant association between the high expression level of RGS5 and low serum calcium level and elevated white blood cells level.

Comments :

Although written well and scientifically interesting, my main concerns are with the figures and conclusions of the document. In general, all of the captions of the figures need to contain more information. Also, I find the coloring in the figures to be confusing and sometimes misleading.

1. Figure 1. A legend/explanation needs to explain the significance between the orange and blue colors. Also, are these individuals or average expression levels?
2. Figure 2 (left) I interpreted the figure to contain data from cases of Grades 1-4 of RCC. So, "Tumor free" in this context confuses me. Also, the significance shown in this figure (and all figures) is between all of the data shown? Or only the "with tumor" category?
3. Figure 2 (right). Is there any interpretable meaning/conclusion between the separation of TNM statuses (e.g. T2, T2a, T2b)? This is again a case where the multitude of colors/categories seems to not add to the interpretation of the figure.
4. Line 43 on Page 3. The authors mention no difference between RGS5 expression of female patients with a primary tumor in the left kidney. This result is mentioned a couple of times in the manuscript and could use some further insight/discussion/hypothesis as to why this might be the case. If the authors feel this is a meaningful conclusion, then further explanation would strengthen this claim.
5. Figure 4. Similar to Figure 2. What does "tumor free" mean? It seems to mean something different in the left versus right figures. The left figure seems to use "tumor free" to mean someone without a tumor? While in the right figure seems to imply someone with a bilateral difference in tumor statuses? (e.g. Tumor present in the right kidney, but not the left?). More thorough captions would aid in this. Also, the coloring here is again confusing. It would be easier to interpret figure 4 (right) if it was presented along with Figure 2 (left).
6. In general, I would encourage the authors to think very carefully about the coloring schemes. With mixed colors in multiple figures, the meaning becomes lost. I would suggest to group similar colorings together (male/female, left/right) or to avoid coloring them at all.
7. There is a missing reference on like 52 of page 7.
8. The discussion re-reports the results and lacks meaningful conclusions/insights into the interpretation of the results. This needs to be heavily revised to provide context to the work done.

Decision letter (RSOS-191422.R0)

17-Jan-2020

Dear Dr Shahriyari,

The editors assigned to your paper ("RGS5 Plays a Significant Role in Renal Cell Carcinoma") have now received comments from reviewers. We would like you to revise your paper in accordance with the referee and Associate Editor suggestions which can be found below (not

including confidential reports to the Editor). Please note this decision does not guarantee eventual acceptance.

Please submit a copy of your revised paper before 09-Feb-2020. Please note that the revision deadline will expire at 00.00am on this date. If we do not hear from you within this time then it will be assumed that the paper has been withdrawn. In exceptional circumstances, extensions may be possible if agreed with the Editorial Office in advance. We do not allow multiple rounds of revision so we urge you to make every effort to fully address all of the comments at this stage. If deemed necessary by the Editors, your manuscript will be sent back to one or more of the original reviewers for assessment. If the original reviewers are not available, we may invite new reviewers.

- Data accessibility

If you wish to submit your supporting data or code to Dryad (<http://datadryad.org/>), or modify your current submission to dryad, please use the following link:
<http://datadryad.org/submit?journalID=RSOS&manu=RSOS-191422>

- Competing interests

- Authors' contributions

All submissions, other than those with a single author, must include an Authors' Contributions section which individually lists the specific contribution of each author. The list of Authors should meet all of the following criteria; 1) substantial contributions to conception and design, or

acquisition of data, or analysis and interpretation of data; 2) drafting the article or revising it critically for important intellectual content; and 3) final approval of the version to be published.

- Acknowledgements

- Funding statement

on behalf of Prof Steve Brown (Subject Editor)
openscience@royalsociety.org

Associate Editor's comments:

Please accept our apologies for the delay in reviewing your paper - we have had to invite an unusually large number of reviewers, and in the event only received one completed report on the manuscript. To avoid delaying you further, we've taken the decision to move forward with the one report.

Please note that you must fully satisfy not only the reviewer but the Editors that your manuscript is ready for acceptance if you choose to revise it, as no further round of revision will likely be possible.

Comments to Author:

Reviewers' Comments to Author:

Reviewer: 1

Comments to the Author(s)

Summary :

In this work, the authors analyzed gene expression profiles of patients with renal cell carcinoma. They found that RGS5 might play a crucial role in initiation and progression of renal cell carcinoma. They have found that when the grade of tumor increases, the RGS5 expression level significantly decrease. They also found that high expression levels of RGS5 are associated with better survival amongst patients. The authors then further investigate this data when separating male and female patients as well as left versus right primary tumors. Finally, they observed a

significant association between the high expression level of RGS5 and low serum calcium level and elevated white blood cells level.

Comments :

Although written well and scientifically interesting, my main concerns are with the figures and conclusions of the document. In general, all of the captions of the figures need to contain more information. Also, I find the coloring in the figures to be confusing and sometimes misleading.

1. Figure 1. A legend/explanation needs to explain the significance between the orange and blue colors. Also, are these individuals or average expression levels?
2. Figure 2 (left) I interpreted the figure to contain data from cases of Grades 1-4 of RCC. So, "Tumor free" in this context confuses me. Also, the significance shown in this figure (and all figures) is between all of the data shown? Or only the "with tumor" category?
3. Figure 2 (right). Is there any interpretable meaning/conclusion between the separation of TNM statuses (e.g. T2, T2a, T2b)? This is again a case where the multitude of colors/categories seems to not add to the interpretation of the figure.
4. Line 43 on Page 3. The authors mention no difference between RGS5 expression of female patients with a primary tumor in the left kidney. This result is mentioned a couple of times in the manuscript and could use some further insight/discussion/hypothesis as to why this might be the case. If the authors feel this is a meaningful conclusion, then further explanation would strengthen this claim.
5. Figure 4. Similar to Figure 2. What does "tumor free" mean? It seems to mean something different in the left versus right figures. The left figure seems to use "tumor free" to mean someone without a tumor? While in the right figure seems to imply someone with a bilateral difference in tumor statuses? (e.g. Tumor present in the right kidney, but not the left?). More thorough captions would aid in this. Also, the coloring here is again confusing. It would be easier to interpret figure 4 (right) if it was presented along with Figure 2 (left).
6. In general, I would encourage the authors to think very carefully about the coloring schemes. With mixed colors in multiple figures, the meaning becomes lost. I would suggest to group similar colorings together (male/female, left/right) or to avoid coloring them at all.
7. There is a missing reference on like 52 of page 7.
8. The discussion re-reports the results and lacks meaningful conclusions/insights into the interpretation of the results. This needs to be heavily revised to provide context to the work done.

Author's Response to Decision Letter for (RSOS-191422.R0)

See Appendix A.

RSOS-191422.R1 (Revision)

Review form: Reviewer 1

Is the manuscript scientifically sound in its present form?

Yes

Are the interpretations and conclusions justified by the results?

Yes

Is the language acceptable?

Yes

Do you have any ethical concerns with this paper?

No

Have you any concerns about statistical analyses in this paper?

No

Recommendation?

Accept as is

Comments to the Author(s)

The clarity of the manuscript is greatly improved by the work the authors have done on the figures and discussion. I recommend that the manuscript be accepted for publication.

Decision letter (RSOS-191422.R1)

31-Mar-2020

Dear Dr Shahriyari,

It is a pleasure to accept your manuscript entitled "RGS5 Plays a Significant Role in Renal Cell Carcinoma" in its current form for publication in Royal Society Open Science. The comments of the reviewer(s) who reviewed your manuscript are included at the foot of this letter.

Please ensure that you send to the editorial office an editable version of your accepted manuscript, and individual files for each figure and table included in your manuscript. You can send these in a zip folder if more convenient. Failure to provide these files may delay the processing of your proof.

Kind regards,

Andrew Dunn

on behalf of Prof Steve Brown (Subject Editor)

Reviewer comments to Author:

Reviewer: 1

Comments to the Author(s)

The clarity of the manuscript is greatly improved by the work the authors have done on the figures and discussion. I recommend that the manuscript be accepted for publication.

Appendix A

Response to the comments for the manuscript entitled "RGS5 Plays a Significant Role in Renal Cell Carcinoma"

Sumeyye Su and Leili Shahriyari
Manuscript ID RSOS-191422

Reviewers' Comments to Author:

Reviewer: 1

Comments to the Author(s)

Summary :

In this work, the authors analyzed gene expression profiles of patients with renal cell carcinoma. They found that RGS5 might play a crucial role in initiation and progression of renal cell carcinoma. They have found that when the grade of tumor increases, the RGS5 expression level significantly decrease. They also found that high expression levels of RGS5 are associated with better survival amongst patients. The authors then further investigate this data when separating male and female patients as well as left versus right primary tumors. Finally, they observed a significant association between the high expression level of RGS5 and low serum calcium level and elevated white blood cells level.

Comments :

Although written well and scientifically interesting, my main concerns are with the figures and conclusions of the document. In general, all of the captions of the figures need to contain more information. Also, I find the coloring in the figures to be confusing and sometimes misleading.

We would like to thank the reviewer for pointing out these important issues; we improved the coloring in the figures, the captions, and the Discussion section. We are very grateful to the reviewer for the great suggestions, which led to a much stronger manuscript. In order to show clearly where changes have been made to the manuscript, we used blue color in the PDF file of the manuscript, which has been attached to this letter.

1. Figure 1. A legend/explanation needs to explain the significance between the orange and blue colors. Also, are these individuals or average expression levels? We edited the caption of this Figure to clearly explain each blue dot represents the average of the expression levels and the orange dots show the average plus standard deviation. The new caption is:

Sub-figure A shows the average expression levels of each gene across patients by blue dots and the corresponding average plus standard deviation by orange dots. Sub-figure B shows the average of gene expression profile of a primary tumor by blue dots, and the corresponding average plus standard deviation by orange dots.

2. Figure 2 (left) I interpreted the figure to contain data from cases of Grades 1-4 of RCC. So, "Tumor free" in this context confuses me. Also, the significance shown in this figure (and all figures) is between all of the data shown? Or only the "with tumor" category?

We edited the colors of this Figure. Since we have discussed the "tumor free" and "with tumor" later in the manuscript, to avoid confusion we removed the color code of tumor status in this Figure. The new caption is:

Expression level of the RGS5 gene as a function of grade and TNM staging of RCC. The sub-figure A shows that the expression level of RGS5 in the patients' primary tumor decreases when the grade of tumor increases. The sub-figure B represents the expression level of RGS5 in primary tumors categorized based on TNM stage of RCC. For some patients with the stage Ti (green), i=1, 2, 3, 4 cancer, we know if the patient had the stage Tia (light green) or Tib (dark green) cancer, for more information see Table 1. Tables in this figure indicate the number of patients in each category.

3. Figure 2 (right). Is there any interpretable meaning/conclusion between the separation of TNM statuses (e.g. T2, T2a, T2b)? This is again a case where the multitude of colors/categories seems to not add to the interpretation of the figure.

We edited this sub-figure too, and we used a new color code. We have explained in the caption of this figure that for some patients with the stage Ti (green), i=1, 2, 3, 4 cancer, we know if the patient had the stage Tia (light green) or Tib (dark green) cancer, for more information see Table 1.

4. Line 43 on Page 3. The authors mention no difference between RGS5 expression of female patients with a primary tumor in the left kidney. This result is mentioned a couple of times in the manuscript and could use some further insight/discussion/hypothesis as to why this might be the case. If the authors feel this is a meaningful conclusion, then further explanation would strengthen this claim.

Since this sentence, "However, there is no significant difference between RGS5 expression of the female patients with a primary tumor in the left kidney compared to the right kidney (Figure 3).", was referenced to the Figure 3, we added the following sentences to the caption of Figure 3:

Sub-figure A shows that female patients have higher expression level of RGS5 than male patients. Grouping patients based on the location of their primary tumors (sub-figure B) shows that the difference between the expression level of RGS5 in female and male patients is due to the fact that male patients with primary tumors in the left kidney have lower expression level of RGS5 than other patients. Tables in this figure indicate the number of patients in each category.

We also added the following paragraph in the Discussion section:

Several studies show that men with RCC have higher stage and grade and ultimately worse overall survival than women with RCC [3,33]. We also observed that the survived female patients have a significantly higher expression level of RGS5 compared to other categories (Figure 5). Moreover, some studies reported better clinical results such as low grade tumors and better survival in patients with right-sided RCC compared to the left-sided RCC [34,35], but our result shows no significant difference between the expression level of RGS5 in female versus male patients with primary tumors located in the right kidney. However, among patients with primary tumors in the left kidney, female patients have a significantly higher RGS5 expression level (Figure4).

5. Figure 4. Similar to Figure 2. What does “tumor free” mean? It seems to mean something different in the left versus right figures. The left figure seems to use “tumor free” to mean someone without a tumor? While in the right figure seems to imply someone with a bilateral difference in tumor statuses? (e.g. Tumor present in the right kidney, but not the left?). More thorough captions would aid in this. Also, the coloring here is again confusing. It would be easier to interpret figure 4 (right) if it was presented along with Figure 2 (left).

We changed the color coding that was very confusing, and we only used colors to show the gender of the patient. We also updated the caption. In the entire paper, “Tumor Free” means that the patient does not have any tumor at the time of the last follow up. We added the following sentence in the results section:

If patients did not have any tumor at the last time of follow up, they were categorized as “tumor free”, and if they had any tumor then they were marked as “with tumor”.

6. In general, I would encourage the authors to think very carefully about the coloring schemes. With mixed colors in multiple figures, the meaning becomes lost. I would suggest to group similar colorings together (male/female, left/right) or to avoid coloring them at all.

We totally agree with reviewer, so we changed the coloring of almost all plots. We now mostly used colors to show the gender. Now, in the entire paper, males are shown by blue color and females by red color.

7. There is a missing reference on like 52 of page 7.

Corrected.

8. The discussion re-reports the results and lacks meaningful conclusions/insights into the interpretation of the results. This needs to be heavily revised to provide context to the work done.

We totally agree, and we rewrote the Discussion and added more references.